# Enhancing the resilience to flooding induced by levee breaches in lowland areas: a methodology based on numerical modelling

Alessia Ferrari[1], Susanna Dazzi[1], Renato Vacondio[1], and Paolo Mignosa[1]

[1]Department of Engineering and Architecture, University of Parma, Parco Area delle Scienze 181/A, 43124 Parma, Italy

**Correspondence:** Alessia Ferrari (alessia.ferrari@unipr.it)

**Abstract.** With the aim of improving resilience to flooding and increasing preparedness to face levee breach-induced inundations, this paper presents a methodology to create a wide database of numerically simulated flooding scenarios due to embankment failures, applicable to any lowland area protected by river levees. The analysis of the detailed spatial and temporal flood data obtained from these hypothetical scenarios is expected to contribute both to the development of civil protection planning, and to immediate actions during a possible future flood event (comparable to one of the available simulations in the database), for which real-time modelling may not be feasible. The most relevant criteria concerning the choice of mathematical model, grid resolution, hydrological conditions, breach parameters and locations are discussed in detail. The proposed methodology, named RESILIENCE, is applied to a 1,100 $km^2$-pilot area in Northern Italy. The creation of a wide database for the study area is made possible thanks to the adoption of a GPU-accelerated shallow water numerical model, which guarantees a remarkable computational efficiency (ratios of physical to computational time up to 80) even for high-resolution meshes (2.5-5 m) and very large domains (> 1,000 $km^2$).

## 1 Introduction

Flood events adversely affect communities living in flood-prone areas, causing huge damages in terms of economic losses and human lives. Recent studies identified a rising trend in flood frequency and affected population in the past few decades (Kundzewicz et al. (2013), Paprotny et al. (2018)), and suggested that global warming will determine a growing occurrence of extreme flood events (Alfieri et al. (2015)) and the related damages (Dottori et al. (2018)) in the future.

Among the possible causes of flooding, levee breaching deserves special attention. Due to the well-known "levee-effect", structural flood protection systems, such as levees, determine an increase in flood exposure. In fact, the presence of this hydraulic defence creates a feeling of safety among people living in flood-prone areas, resulting in the growing of settlements and in the reduction of preparedness, hence in the increase of vulnerability in those areas (Di Baldassarre et al. (2015)). As a result, more people are exposed to less frequent but more devastating floods, for which the statistical frequency is difficult to assess, due to the historical changes in river systems (Black (2008)).

Moreover, the presence of levees causes a reduced flood attenuation, which in turn increases the damages when a breach occurs (Heine and Pinter (2012)). Despite all the efforts adopted in embankment design, maintenance and monitoring, the residual risk associated with levee breach flooding in the surrounding areas cannot be neglected, and its evaluation is hence gaining attention worldwide (Huthoff et al. (2015), Pinter et al. (2016)). Nowadays, mathematical models for flood simulation, which solve physically-based equations for hydrodynamics (Teng et al. (2017)), represent an essential instrument for flood hazard and risk assessment (e.g. Apel et al. (2004), Qi and Altinakar (2011)), including the residual risk due to levee breaches (Vorogushyn et al. (2010)). Numerical modelling can contribute to draw up flood risk management plans with prevention and protection measures to reduce flood-related damages and enhance resilience (defined as the ability of the system to recover from flooding in flood-prone areas, De Bruijn (2005)). The accurate predictions of inundation scenarios provided by numerical simulations can also be useful for assessing civil protection and adaptation strategies (Jongman et al. (2018)) and emergency planning during flood events (Tarrant et al. (2005), Dulebenets et al. (2019a), Dulebenets et al. (2019b)).

For civil protection purposes, real-time numerical modelling is the most suited solution when dealing with large river basins whose flood events last several days, considering that simulation time (a few hours) is usually much smaller than the physical duration of the flood. Moreover, hydrologic inputs or water level measurements from upstream river sections can be used as boundary conditions to predict the flood propagation along the river. Conversely, in small/medium river basins with short-lasting floods (less than one day), real-time simulations are much more challenging, because: (i) they have to be based on rainfall-runoff models and weather forecast, which are characterized by high uncertainties, and (ii) their computational and physical times are characterized by the same order of magnitude. Focusing on levee failures, the real-time prediction of possible breach locations is very difficult in the practice, due to the complexity of the breaching process and to the uncertainties in the embankment material characteristics (often heterogeneous, and with unknown local discontinuities), especially for small rivers. Considering all these limitations, the creation of an off-line database of hypothetical flooding scenarios constitutes an alternative solution to real-time forecasting based on integrated hydrologic-hydraulic modelling.

This paper presents a methodology for assessing the flooding scenarios induced by levee breaches with the purpose of increasing resilience in lowland areas. For a given exposed area, the RESILIENCE project (REsearch on Scenarios of Inundation of Lowlands Induced by EmbaNkment Collapses in Europe) aims at the creation of a wide database of high-resolution numerical simulations, concerning several hypothetical flood scenarios, each one characterized by a specific breach location and an upstream discharge hydrograph with assigned return period. While previous studies combined the results of different scenarios in order to create probabilistic flood hazard and flood risk maps (Di Baldassarre et al. (2009), Vorogushyn et al. (2010)), in this work breach scenarios are not associated with their probability of occurrence. In fact, the focus of this study is not on flood hazard mapping, but on the evaluation of flood dynamics, arrival times, maximum water depths and velocities, required for the definition of civil protection strategies, which should be equally effective regardless of the event probability, breach failure mechanism, etc.. Accurate simulation results are obtained thanks to the adoption of high-resolution meshes and of a robust and efficient numerical model, named PARFLOOD (Vacondio et al. (2014), Vacondio et al. (2017)).

The methodology is applied to a study area in Northern Italy, bounded by the Po River and by its two tributaries Secchia and Panaro, which was affected by levee breach flooding in the past (Vacondio et al. (2016)). General guidelines for the application

of the procedure and details on the criteria adopted for the pilot area are reported. Moreover, a few examples of simulation results are provided, and their possible practical use is discussed.

The paper is organized as follows: in Sect. 2, the RESILIENCE project is presented, and the most important features and requirements of the methodology are described in detail. Sect. 3 illustrates the application to the pilot area, together with some examples of the results. The assumptions, the advantages and the implications of the methodology are discussed in Sect. 4, and concluding remarks are finally outlined in Sect. 5.

## 2 Flooding scenarios induced by levee breaches: the RESILIENCE project

The RESILIENCE project aims at defining a new methodology for mapping flood scenarios due to levee breaches, which can be helpful for improving preparedness and supporting the development of technical and scientific tools for emergency planning and management, consistently with EU Flood Directive 2007/60/CE. Several breach locations along a leveed river are preliminarily identified, and multiple discharge hydrographs, characterized by different return period, are considered. Each combination of breach position and upstream boundary condition corresponds to a simulated flood scenario. In this way, a large database describing different hypothetical real levee breach events in that area is created. The results of these simulations, made available to public administrations, can be fundamental not only for emergency planning, but also for taking appropriate actions of civil protection in the course of real flood events.

In the following sections, the most important assumptions of the methodology concerning model selection, spatial resolution, hydrological conditions, breach locations and modelling are discussed thoroughly. Moreover, the most relevant simulation outputs and their usefulness for civil protection purposes are described.

### 2.1 Numerical model

#### 2.1.1 Background

Free-surface flows are traditionally described by means of the Shallow Water Equations (SWEs), i.e. depth-averaged mass and momentum conservation laws (Toro (2001)), which can be written either in one-dimensional (1D) or in 2D form. In the past, the high computational effort required to perform fully 2D simulations led to the development of 1D-2D models, which separate the river, described by means of a 1D model, and the flood-prone area, where a 2D model is adopted because in this region no preferential flow direction can be determined *a priori*.

However, the adoption of either 1D-2D uncoupled (Di Baldassarre et al. (2009), Masoero et al. (2013), Mazzoleni et al. (2013)) or coupled (e.g. Gejadze and Monnier (2007), Morales-Hernández et al. (2013), Bladé et al. (2012)) models may lead to inaccurate results. In the former case, backwater effects near the breach location, which can reduce the outflow discharge or even reverse the flow (Viero et al. (2013)), are not captured, whereas in the latter case the need of defining the coupling location *a priori* makes the 1D-2D model less flexible than a fully 2D model. Besides, the flow field becomes markedly 2D after the breach opening, both inside and outside the river region, and a 1D model cannot predict the outflowing discharge accurately.

Focusing on the 2D-SWEs, several models adopt a diffusive approach to simplify the original formulation (e.g. LISFLOOD, Horritt and Bates (2002)). However, this does not always guarantee an overall accuracy comparable to that obtained from models which solve the full SWEs, particularly when supercritical flows and hydraulic jumps need to be modelled (Hunter et al. (2008), Neal et al. (2012), Costabile et al. (2017)), as often occurs when a breach opens. On the other hand, if the complete equations are written in conservative form, explicit finite volume (FV) schemes can be adopted (Toro (2001)). These methods have the advantage of reproducing both subcritical and supercritical flows, of incorporating the propagation of shock-type discontinuities automatically, and of including robust treatments of wet/dry fronts and irregular topography (Liang and Marche (2009)).

The high computational cost of fully 2D models based on complete SWEs has prevented their extensive use in simulating large domains ($> 100$ km$^2$) with high resolution meshes (5-10 m) until a few years ago, when parallelization techniques started being applied to SWE models (Sanders et al. (2010)). Nowadays, expensive supercomputers are not even required, since the use of Graphics Processing Units (GPUs) (Brodtkorb et al. (2012), Lacasta et al. (2014), Vacondio et al. (2014), Vacondio et al. (2017), Conde et al. (2017)) limits hardware requirements to a standard workstation equipped with a video card. The execution time of a GPU-parallelized code can be reduced up to two orders of magnitude compared to the CPU serial version of the same code (Castro et al. (2011), Vacondio et al. (2014), García-Feal et al. (2018)).

### 2.1.2 The PARFLOOD model

Following this discussion, a GPU-accelerated fully 2D model, such as PARFLOOD (Vacondio et al. (2014), Vacondio et al. (2017)), is considered best suited for the purposes of this work and is adopted for the present application. The model solves the complete 2D SWEs written in integral form (Toro (2001)):

$$\frac{\partial}{\partial t} \int_A \mathbf{U} dA + \int_C \mathbf{H} \cdot \mathbf{n} dC = \int_A \left( \mathbf{S}_0 + \mathbf{S}_f \right) dA \tag{1}$$

where $t$ is the time, $A$ and $C$ are the area and boundary of the integration volume, respectively, $\mathbf{U}$ is the vector of conserved variables, $\mathbf{H} = (\mathbf{F},\mathbf{G})$ is the tensor of fluxes in the $x$ and $y$ directions, $\mathbf{n}$ is the outward unit vector normal to $C$, while $\mathbf{S}_0$ and $\mathbf{S}_f$ are the bed slope and friction source terms, respectively. In order to obtain a well-balanced scheme, the terms $\mathbf{U}$, $\mathbf{F}$ and $\mathbf{G}$, $\mathbf{S}_0$ and $\mathbf{S}_f$ are defined according to the formulation of Liang and Marche (2009), as follows:

$$\mathbf{U} = \begin{bmatrix} \eta \\ uh \\ vh \end{bmatrix}, \mathbf{F} = \begin{bmatrix} uh \\ u^2 h + \frac{1}{2}g(\eta^2 - 2\eta z) \\ uvh \end{bmatrix}, \mathbf{G} = \begin{bmatrix} vh \\ uvh \\ v^2 h + \frac{1}{2}g(\eta^2 - 2\eta z) \end{bmatrix}, \mathbf{S}_0 = \begin{bmatrix} 0 \\ -g\eta\frac{\partial z}{\partial x} \\ -g\eta\frac{\partial z}{\partial y} \end{bmatrix}, \mathbf{S}_f = \begin{bmatrix} 0 \\ -gh\frac{n_f^2 u \sqrt{u^2 + v^2}}{h^{4/3}} \\ -gh\frac{n_f^2 v \sqrt{u^2 + v^2}}{h^{4/3}} \end{bmatrix} \tag{2}$$

where $h$ is the flow depth, $z$ is the bed elevation, and $\eta = h + z$ is the water surface elevation; moreover, $u$ and $v$ are the velocity components along the $x$ and $y$ directions, respectively, $g$ is the acceleration due to gravity, and $n_f$ is Manning's roughness coefficient.

Equations 1 and 2 are discretized using an explicit FV scheme, and both first- and second-order accurate approximations in space and time can be selected in the PARFLOOD model. The adoption of a depth-positive MUSCL extrapolation at cell boundaries (Audusse et al. (2004)) with the *minmod* slope limiter, and the second-order Runge-Kutta method ensures the second-order accuracy in space and time, respectively. Flux computation is performed using the HLLC approximate Riemann solver (Toro (2001)), and the correction proposed by Kurganov and Petrova (2007) to avoid non-physical velocity values at wet/dry fronts is applied. Finally, the slope source tem is discretized by means of a centered approximation (Vacondio et al. (2014)), while for the friction source term the implicit formulation proposed by Caleffi et al. (2003) is adopted. The computational grid can be either Cartesian or Block Uniform Quadtree (BUQ, see Vacondio et al. (2017)), that is a non-uniform structured mesh.

With the aim of reducing the computational times, the code is written in Compute Unified Device Architecture (CUDA) language, i.e. a framework for GPU-based parallel computing introduced by NVIDIA[TM]. This high-level language allows for the exploitation of both hardware resources: the CPU (the host) and the GPU (the device). The good computational performance of this code for field applications was assessed in previous works (Vacondio et al. (2016), Dazzi et al. (2018), Dazzi et al. (2019), Ferrari et al. (2018), Ferrari et al. (2019)).

## 2.2 Topographic data and spatial resolution

When levee breach-induced floods in lowland areas are modelled, a high-resolution mesh is often necessary to describe the relevant terrain features typical of man-made landscapes (e.g. roads, railways, channels, embankments). High-resolution topographic information for large areas can be obtained from new remote sensing techniques, such as Light Detection and Ranging (LiDAR) and Shuttle Radar Topography Mission (SRTM), which provide raw data for the generation of digital terrain models (DTMs). LiDAR-based DTMs are nowadays available to public access for most flood-prone areas in Europe, and meshes derived from these data (even coarsened) often provide the most accurate results for flood modelling (Ali et al. (2015)).

DTM grids can include billions of cells; however, the amount of cells (and thus the runtimes) can be decreased by performing a moderate downsampling (e.g. reducing the grid size from 0.5-1 m to 2-5 m) and by adopting non-uniform meshes, either unstructured (Liang et al. (2008), Sætra et al. (2015)) or structured (Vacondio et al. (2017)). In fact, in urban and suburban areas, the presence of road and railway embankments can influence the flood dynamics significantly, and the bathymetry near these elements should be at high resolution (2-5 m). On the other hand, for uniform rural areas a lower resolution (e.g. 10-50 m) can be used without impairing the overall accuracy.

It is relevant to notice that high-resolution DTMs can be exploited thanks to the availability of parallel 2D codes; until a few years ago, traditional 2D and 1D-2D models usually adopted a low resolution in the order of 50-100 m for flood-prone areas in order to reduce the computational times (Aureli and Mignosa (2004), Aureli et al. (2005), Vorogushyn et al. (2010), Masoero et al. (2013), Mazzoleni et al. (2013), Huthoff et al. (2015)).

## 2.3 Upstream/downstream boundary conditions

Discharge hydrographs with a specific return period are assigned as upstream boundary condition. Sometimes these hydrographs are already available from previous hydrological studies, and can be provided by local River Basin Authorities; otherwise, they can be derived from rainfall-runoff modelling or from statistical analyses of recorded discharge hydrographs (e.g. Tomirotti and Mignosa (2017)).

For the purpose of this study, multiple hydrological conditions should be considered, in order to cover possible configurations characterized by different breach triggering mechanisms, flood volumes, etc. At least two different discharge hydrographs should be considered for each breach location, even though the simulation database can be extended with more hydrological inputs if needed. The first case ("inflow A") corresponds to the condition for which the water surface elevation reaches the levee crest somewhere along the river, thus generating overtopping. The second configuration ("inflow B") concerns a flood event with a lower return period, for which the levee is never overtopped, but other mechanisms might induce the levee collapse. In fact, earthen levees can also experience breaching for piping and internal erosion processes, even when water levels remain below the levee crest. Besides, the dens of burrowing animals (e.g. porcupine, badger, nutria) were recently identified as another possible cause for breach triggering (Viero et al. (2013), Orlandini et al. (2015)). Incidentally, the collapse of an embankment during a flood event with a relatively low return period can be very threatening for human lives because the highest warning thresholds may not be reached, and population can be unprepared to face flooding.

The choice of the return period for inflows A and B is river-dependent, because it is influenced by the design return period of the levee system, by the presence of other flood control structures, etc. In general, preliminary simulations of the propagation of flood waves with different return periods (e.g. 10, 20, 50, 100, 200, 500 years) for each river should be performed, and the event that corresponds to incipient overtopping can be identified as inflow A. Then, a higher frequency hydrograph can be selected as inflow B, in order to consider levee collapses due to piping or other mechanisms during a flood event that does not induce overtopping (for example, when a specific freeboard is guaranteed).

The discharge hydrographs thus obtained are imposed as upstream boundary condition for the levee breach scenarios. The downstream boundary condition also deserves special attention. In fact, downstream of the breach location, the discharge may reduce or even fall to zero and reverse, but the water depth does not necessarily reduce accordingly. Hydraulic gradient and inertia can play a significant role, and the stage-discharge relationships at downstream sections may be characterized by a strong loop effect (D'Oria et al. (2015)). This leads to the suggestion that, if a single-valued rating curve is imposed as outflow boundary condition, it should be assigned at the farthest possible section downstream from the breach location (even if some criteria concerning the optimal location of a downstream unique rating curve were studied, e.g. Singh et al. (1997)).

## 2.4 Levee breaches location and modelling

Several breach locations must be identified along the river levees, so that a real event can be associated with one of the simulated scenarios. The distance between two consecutive breach positions should be selected considering the river dimensions,

the presence of urban settlements, the flood-prone area topography, and the possible presence of roads and embankments influencing the inundation dynamics.

The description of the gradual opening of the levee breach must be somehow included in the 2D modelling, since the hypothesis of instantaneous break is not realistic for river embankments. Among the available approaches in the literature, which also include the coupling of the SWEs with a sediment transport model (Faeh (2007)), or with an erosion law (Dazzi et al. (2019)), the simple "geometric" approach is selected in this work for two reasons. First, the uncertainties about the geotechnical parameters of the embankment and the complexity of the breaching process (three-dimensionality, interactions between erosion, infiltration, and bank stability, etc.) can be neglected. Second, this method was successfully applied to a real test case (Vacondio et al. (2016)), and a similar approach is often adopted for 1D-2D coupled models, especially in the context of flood hazard assessment (e.g. Vorogushyn et al. (2010), Mazzoleni et al. (2013)). Thus, in the RESILIENCE project the breach evolution is modelled by varying the local topography in time assuming a trapezoidal opening that widens and deepens from the crest of the embankment to the ground elevation; the breach geometric dimensions (i.e. width) and opening time are given as input parameters.

The two breach parameters should be set consistently with historical data, if available (e.g. Nagy (2006), Vorogushyn et al. (2010), Govi and Maraga (2005)), or hypothesized according to the river and embankment characteristics. A breach final width in the order of tens to hundreds of meters is often assumed in the literature (Apel et al. (2006), Kamrath et al. (2006)). As for the breach development time, very few field observations are available, and values in the range 1-3 h are often reported (Apel et al. (2006), Alkema and Middelkoop (2005)). The impact of the parameters uncertainty on the results should be evaluated for at least one hydrological scenario, by means of a probabilistic treatment (Apel et al. (2006), Vorogushyn et al. (2010), Mazzoleni et al. (2013)) or a sensitivity analysis (Kamrath et al. (2006), Huthoff et al. (2015)).

For each breach location and return period, the triggering time for breach opening should be selected after preliminary simulations and corresponds to the moment when either overtopping or the peak water surface elevation is observed at the breach location.

Finally, simulations must be extended in time long enough to capture the flooding evolution in the lowland, which is actually the goal of the presented methodology. The selection of this temporal interval should be guided by considerations on the flood duration in the river, on the inundation dynamics, and on the fact that drainage operations and/or breach closure would start at some point.

## 2.5 Outputs

The outcomes of the modelled scenarios could help the civil protection activities for emergency planning and/or at the occurrence of an event similar to one of those already modelled. Arrighi et al. (2019) recently presented a framework that integrates hydrologic/hydraulic modelling with human safety and transport accessibility evaluations, applied to a small municipality near Florence (Italy), and confirmed the usefulness of detailed spatial and temporal flood data provided by numerical modelling for civil protection purposes. Thus, the first mandatory output concerns spatial and temporal information about the flood dynamics

in the lowland area, which can be visualized as an animation of the inundation pattern or as a sequence of frames at selected times.

Other useful indicators for evaluating the flood severity for each scenario include the maps of arrival times of the wetting front, maximum water depths (and/or water surface elevations), maximum velocities, and of a hazard index which combines simultaneous water depth and velocity. In general, these maps can be opened in a GIS environment and overlapped with layers of data (e.g. about population, transportation, buildings, critical and vulnerable structures, etc.) to analyse the possible flood impacts on the territory, with the aim of increasing the resilience in the area. First, information about the maximum water depth allows for the definition of the affected areas, as well as the identification of escape routes and safety zones where assembly points can be organized. On the other hand, in areas where only shallow water levels can be expected, people can simply be instructed to protect their homes with anti-flood barriers to prevent water from damaging their property. Besides, analysis of the inundation dynamics can reveal possible service interruptions and disturbance to road accessibility, which can also prevent rescue operations; in this way, alternative routes can be identified.

Moreover, the maximum flow velocity map should not be neglected, because high velocities can reduce the stability of vehicles and pedestrians, increasing the hazard for human lives (Milanesi et al. (2015)). In general, the most important results for quantifying the hazard for human lives are the maximum simultaneous water depth and velocity, which can be represented in terms of Froude number, total head, total force and/or total depth. This last indicator, which represents the water depth at rest $D$ whose static force is equivalent to the total force of the flow, is evaluated as follows (Aureli et al. (2008), Ferrari et al. (2019)):

$$D(t) = h(t)\sqrt{1 + 2Fr^2(t)} \tag{3}$$

where $h(t)$ represents the water depth and $Fr(t)$ the Froude number at time $t$.

Finally, the map of the arrival times of the wetting front can be useful for early warning and for establishing the timeline for the evacuation procedures, in particular for vulnerable people (older adults, hospitalised patients, etc.). The off-line analysis of the simulation results, on the other hand, can help in the identification of possible strategies to reduce the damages, as for example the adoption of movable defence systems (e.g. flood barriers) to preserve lowland urban settlements from flooding or other emergency interventions to drain water (e.g. pumping, relief cuts). Selected strategies can also be tested numerically to verify their effectiveness.

## 3  Application of the RESILIENCE project to a pilot area in Northern Italy

In this section, an example of application of the proposed methodology is presented. The pilot area for the RESILIENCE project (Fig. 1) is at the boundary of two regions (Emilia-Romagna and Lombardia), in Northern Italy. This territorial unit is about 1,100 km$^2$ wide, and is delimited by the Po River (North) and by its two right tributaries Secchia (West) and Panaro (East). The city of Modena bounds the area to the South. This lowland area can be potentially affected by flooding events caused by breaches from the 83 km-long right levee of the Secchia River and/or from the 67 km-long left levee of the Panaro River.

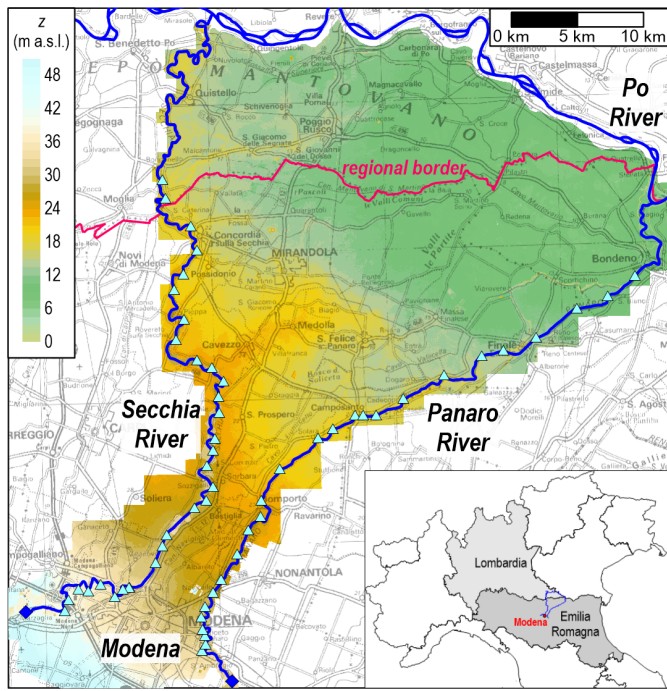

**Figure 1.** Map of the pilot area (Emilia-Romagna and Lombardia regions, Northern Italy): rivers are represented in blue; the breach locations along the Secchia and Panaro Rivers are indicated as triangles in cyan; the terrain elevation contour map is also depicted in background.

This study area was selected for several reasons. First, the latest report of the Italian Institute for Environmental Protection and Research ISPRA (2018) showed that Emilia-Romagna is the Italian region with the highest level of exposed population (up to 2.7 million exposed inhabitants out of 4.3), buildings and areas for both high (return period of 20-50 years) and medium (return period of 100-200 years) flood frequency. Moreover, the middle-lower basin of the Po River was subject to levee breach-induced floods several times in the last 150 years, either from the main river levees or from its leveed tributaries (e.g. Di Baldassarre et al. (2009), Masoero et al. (2013), D'Oria et al. (2015), Dazzi et al. (2019)), often with devastating consequences. Finally, the Secchia and Panaro rivers experienced levee breach events in the past, even without overtopping and during the occurrence of floods with low/medium return periods. In particular, the most recent event that occurred in this area was the flood originated by a bank failure on the Secchia River in 2014 (Vacondio et al. (2016)), which caused roughly 500 million euros losses. This event raised awareness of the huge damages caused by flooding and of the necessity of increasing flood preparedness in both public administrations and population.

## 3.1 Setup

The computational domain was built based on the available 1 m-resolution DTM of the riverbeds and of the floodable lowlands derived from LiDAR surveys carried out in the years 2008, 2015 and 2016. In order to avoid the excessive memory requirements and heavy computational costs (even for a fast GPU-parallel model), related to the adoption of a uniform 1 m mesh (which

would require billions of cells), the DTM was downsampled to a resolution of 5 m. This operation did not affect the crest elevation of the artificial embankments. In fact, each 5×5 m cell crossed by an embankment was identified, and its elevation was set equal to the maximum value among the original 25 points belonging to that cell; otherwise, its elevation was simply computed as the average of the 25 terrain data comprised in that cell. For other urban features that were not captured correctly by the LiDAR survey, additional corrections were introduced manually.

Then, the study domain was discretized by means of a non-uniform BUQ grid: the maximum resolution (5 m) was forced along the riverbed, the levees, the main artificial embankments and channels, and at the breach location, whereas for rural areas it gradually decreased up to 40 m according to the algorithm presented by Vacondio et al. (2017). The resulting computational grid (Fig. 2), whose spatial resolution is considered suitable for the detailed modelling of the river and the lowland area, consists of roughly $13 \cdot 10^6$ cells, and the number of cells is reduced by approximately 70% compared to a uniform 5 m-mesh.

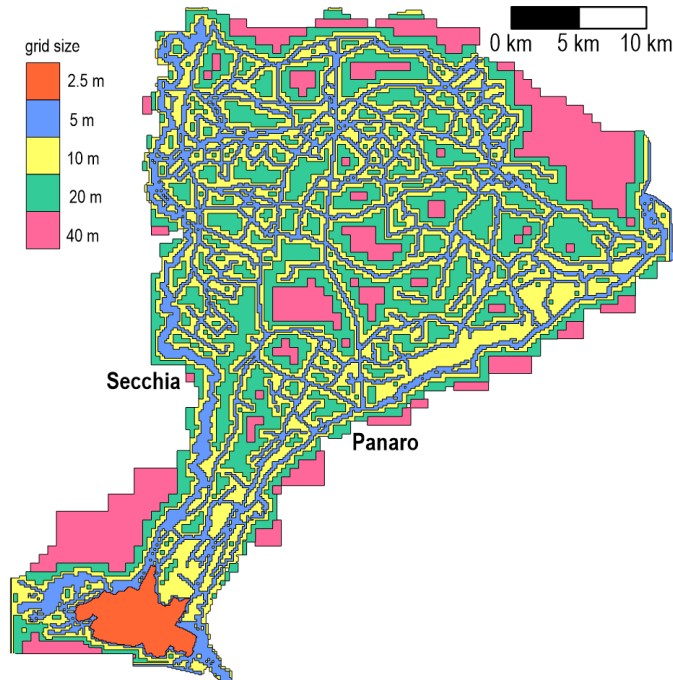

**Figure 2.** Multiresolution computational grid for the pilot area.

The study area also includes several urban settlements: Modena, with about 185,000 inhabitants, is the most populated one, and its old city centre comprises narrow streets, which cannot be accurately described with a 5 m resolution. Therefore, limited to the few scenarios concerning the flooding of Modena, simulations were performed using a finer mesh (up to 2.5 m) in the town, and the urban layout was modelled according to the "building hole" method (Schubert and Sanders (2012)), in order to capture the flow field inside the urban area correctly.

Buildings were explicitly resolved only for the city of Modena, whereas, according to previous findings (Vacondio et al. (2016)), the presence of the other (smaller) settlements was taken into account by means of a higher resistance parameter

("building resistance" method, Schubert and Sanders (2012)). In particular, the roughness coefficient for the urban areas was calibrated based on the event occurred in 2014 (the flood arrival times at selected locations were known), and was set equal to $0.14$ m$^{-1/3}$s, while for rural areas a Manning coefficient of $0.05$ m$^{-1/3}$s was chosen. In the absence of data for calibration concerning past flooding events, land use maps can be exploited to assign standard roughness values from the literature.

As for the river roughness, the calibration for the Secchia River was performed in previous works (Vacondio et al. (2016)), based on the water levels recorded at the available gauging stations during recent flood events. A Manning coefficient equal to $0.05$ m$^{-1/3}$s provided the best results. The roughness of the Panaro River was also subject to calibration with a similar procedure, and the value $0.04$ m$^{-1/3}$s was selected.

As regards the selection of the hydrological scenarios for the two rivers, the Synthetic Design Hydrographs (Tomirotti and Mignosa (2017)) with assigned return periods were considered. After preliminary simulations, inflow A (overtopping-induced flooding) was identified as the 50 years-return period hydrograph for the Secchia River, and as the 100 years-one for the Panaro River. Then, the inflow hydrograph with 20 years-return period was selected as inflow B for both rivers, in order to consider an event with higher frequency. These discharge hydrographs were assumed as upstream boundary conditions during the simulations.

The downstream section was located at the confluence (of the Secchia and Panaro rivers, respectively) into the Po River, and a constant water level in this (much larger) river, which did not affect the breach outflow even for the most downstream breach location, was assumed as boundary condition.

As mentioned before, the pilot area can be flooded by hypothetical failures occurring along the Secchia and Panaro rivers. Assuming a distance of about 2 km from one breach to the other, 30 breach locations were identified along the right levee of the Secchia River (for the flooding scenarios related to this river), and 26 ones along the left levee of the Panaro River. Figure 1 reports all the 56 simulated breach sites. Among these, 8 breach scenarios on the Secchia River and 4 on the Panaro River involve the city of Modena. Based on past observations, the breach final width was assumed equal to 100 m for all scenarios, while the opening time was set equal to 3 or 6 h for inflows A and B, respectively.

All the 112 simulations (56 breach locations and 2 hydrological scenarios) were prolonged for 72 h (3 days), because at that point the outflow from the breach was almost null, the flooded area had reached its maximum extension, and no significant flow dynamics can be observed.

## 3.2 Outcomes

Figure 3 shows an example of the results for one scenario on the Secchia River. The maps of maximum water depths shown in Figure 3a reveals that the flooding involves the northern portion of the domain, partially affecting some urban settlements (S. Possidonio, Mirandola), while villages at east are safe from this inundation and could temporarily host the evacuated population. The maximum velocity magnitude (Fig. 3b) remains below 1 m s$^{-1}$, except for the surroundings of the breach, and close to some road embankments that are overtopped by water (see detail in Fig. 3b), where drivers can be in grave danger. The combination of water depth and velocity (Fig. 3c) highlights that lowland areas are mostly affected with low (green, $0 \leq$ D $<$ 0.5 m) and medium (yellow, $0.5$ m $\leq$ D $<$ 1 m) total depth values, even if higher values (orange, 1 m $\leq$ D $<$ 1.5 m, and red,

D $\geq$ 1.5 m) are reached where high water depths are observed. The map of flood arrival times in Fig. 3d shows that, for this scenario, approximately 45% of the flooded area is affected 24 h after the breach opening (purple contour line), guaranteeing a considerable amount of time for emergency activities. It is relevant to notice that also during the levee breach occurred in 2014 on the Secchia River (Vacondio et al. (2016)) one of the most affected villages was flooded the day after the opening, but no countermeasures were taken at that time, since *a priori* knowledge of the inundation dynamics was not available. Finally, a video showing the flooding evolution for this scenario is provided as additional material.

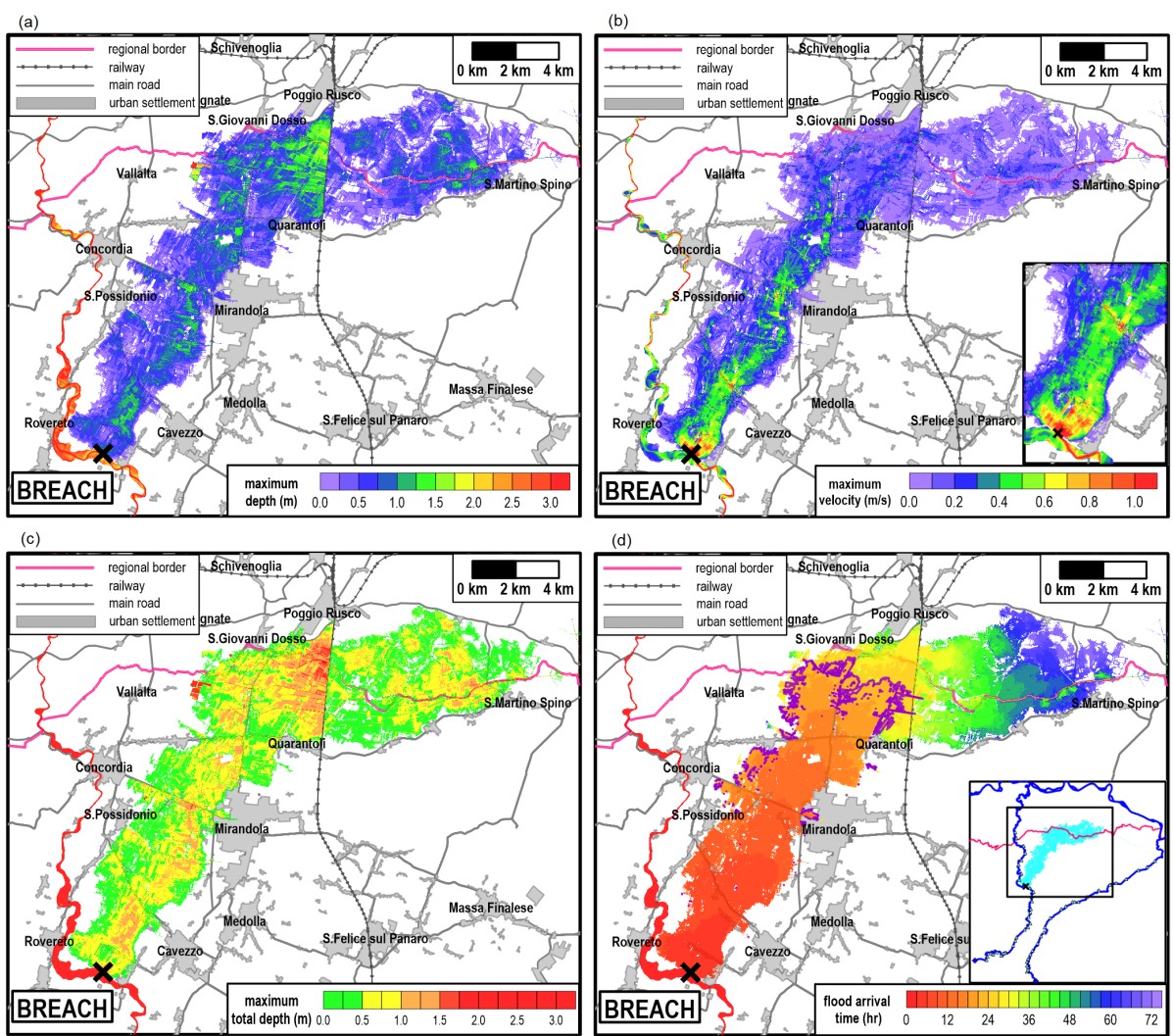

**Figure 3.** Example of the resulting maps concerning the maximum (a) water depth, (b) velocity, (c) total depth and (d) flood arrival time for a given scenario on the Secchia River with inflow B. The main roads, railways, and urban settlements are identified, and the breach location is indicated with a black cross.

In addition to the maps representing specific hydraulic indicators for each scenario, further information can be obtained by analysing the results of the whole database. In particular, the most affected parts in the pilot area and those never hit by flooding can be investigated. Therefore, for both inflows A and B, the maps of the inundated areas were combined in order to quantify the number of scenarios affecting each computational cell in the domain. Figure 4 shows the resulting map for inflow B: in this case, about 50% of the pilot area is affected by at least one breach scenario. In particular, two areas can be identified as most affected by the possible flooding induced by levee breaching, since up to 21 breach scenarios (from both the Secchia and the Panaro River) involve these regions. On the other hand, it can be observed that there is a large zone, in the middle of the pilot area, that is not affected by any of the considered breach scenarios thanks to the favourable terrain topography; hence, it is potentially recommended for evacuation purposes (e.g. organization of assembly points).

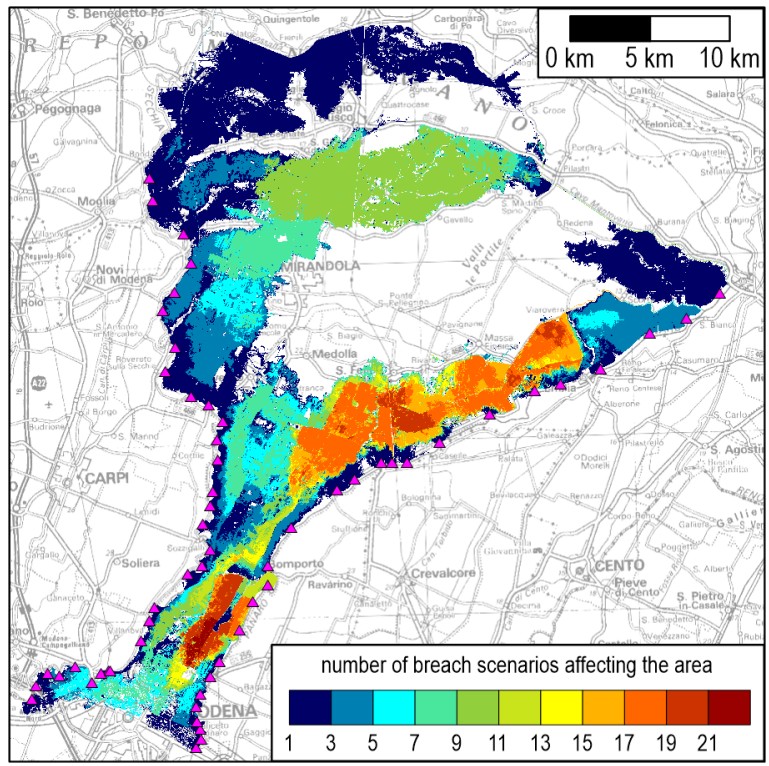

**Figure 4.** Number of flooding scenarios affecting each cell of the pilot area with inflow B. Two portions of the domain (in red) are flooded for 21 scenarios (from either the Secchia or the Panaro River), while the uncoloured zones are never inundated.

## 4 Discussion

Due to climate change and population growth, structural protection systems such as levees have been often adopted to increase protection against floods. However, this kind of defence presents several ecological (e.g. hydraulic decoupling between the river

and its floodplain, loss of biodiversity, change in groundwater levels, increase in greenhouse gas emission) and socio-economic consequences (Auerswald et al. (2019)).

Focusing on the "levee-effect" and on the issue of adequately considering the residual flood risk related to levees, the main goal of this paper was to define a methodology based on the use of numerical models to enhance the resilience of lowland areas in case of levee breach occurrence. The procedure, which requires the numerical simulation of many scenarios with different breach locations and hydrological inputs, is applicable to lowland areas protected from flooding by river levees, which can be inundated in case of embankment collapse.

In the context of emergency planning, the creation of a large database of scenarios represents the main alternative solution when real-time simulations cannot be performed (e.g., when weather forecast systems or direct measurements are missing or simply cannot provide reliable predictions of the incoming flow hydrograph in small river basins). This means that, when a flooding event occurs, the results of the closest simulated scenario can be accessed in order to predict the inundation pattern and to better organize the civil protection activities and take timely countermeasures. Besides these emergencies, the results of each simulation, also combined for global considerations, allow for an improvement of evacuation and defence system planning. In this context, advanced optimization-based tools and algorithms (Dulebenets et al. (2019a), Dulebenets et al. (2019b)) can be exploited to create emergency evacuation plans that efficiently minimize individuals travel time during a natural hazard.

Moreover, the simulation results can be useful for updating the alert systems, as well as for the dissemination of the correct behaviour to local inhabitants. In the framework of adaptation management, recently, the LIFE PRIMES project (LIFE (2019)) contributed to building resilient communities in other areas in the Emilia Romagna region, by raising their awareness and proactive participation in the operations of early warning.

In this framework, the availability of high-resolution DTMs, which can describe the local terrain features in detail, represents a relevant tool. Focusing on numerical modelling, the adoption of a fully 2D-SWE model was claimed not only for capturing the complex hydrodynamic field near the breach, but also the wet/dry fronts propagation over an irregular topography. The only drawback of this kind of models, which is the long computational time, was overcome by taking advantage of a parallelized code, such as PARFLOOD. Simulations were performed using a NVIDIA® Tesla® P100 GPU. Runtimes range approximately between 1 and 5 h, depending on the extent of the flooded area. The ratio of physical time to computational time is between 15 and 80 and confirms the high efficiency of GPU-accelerated codes for flood simulation, even for large high-resolution domains. If HPC clusters equipped with 20 to 50 GPUs could be exploited, the simulation of the whole database of 112 scenarios would only require 18 to 9 h of computation, assuming an average runtime of 3 h.

With reference to the assessment of flooding scenarios involving urban settlements, the use of a fully 2D model and a high-resolution mesh is required. In particular, a grid size in the order of 2-3 m becomes crucial when dealing with historical towns. Evidence of this requirement is shown in Fig. 5 as regards the potential flooding of the city of Modena, whose urban layout was modelled with a 2.5 m-resolution mesh using the "building hole" approach. The backwater effect caused by buildings and the high flow velocities (>1 m s$^{-1}$) along some streets can be observed. Near the historical centre, streets are very narrow, and the complex hydrodynamic field could hardly be captured using a coarser mesh.

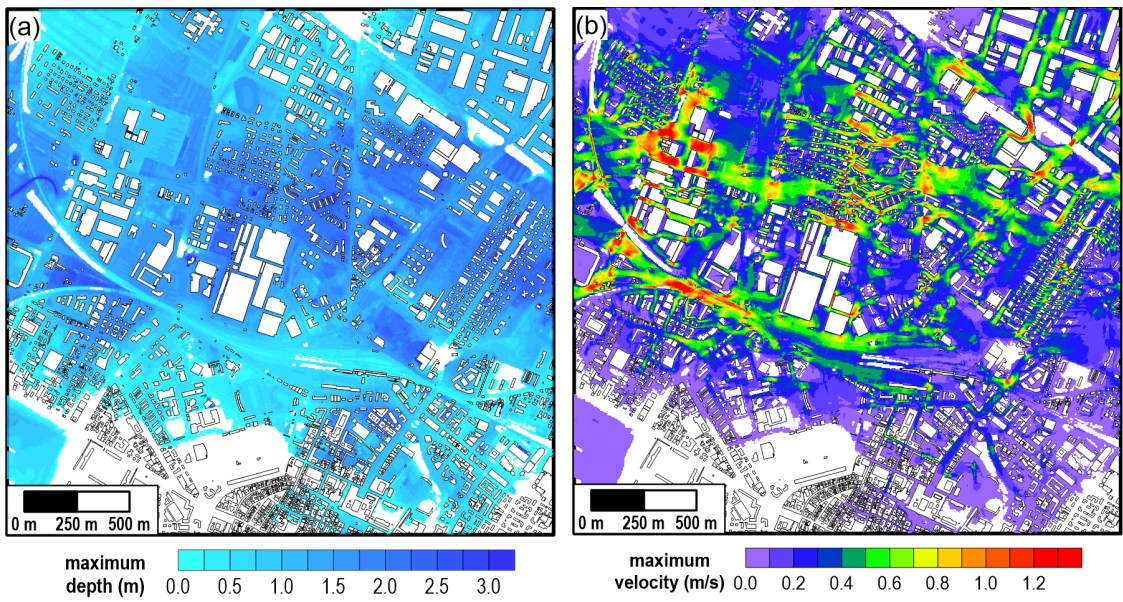

**Figure 5.** Detail of the complex hydrodynamic field in the city of Modena: maximum water (a) depth and (b) velocity for one breach scenario on the Secchia River.

Still focusing on topographic information and spatial resolution, it is relevant to notice that drainage networks with relatively narrow channels were not described in detail in the computational grid. In fact, even though microtopography (i.e. tillage feature, ditches) determines preferential pathways for very shallow flows (Viero and Valipour (2017), Hailemariam et al. (2014)), the maximum discharge through the breaches here considered (in the order of $10^2$ m$^3$/s) largely exceeds the discharge capacity of the drainage systems. Moreover, most of the minor channels are equipped with sluice gates that are kept closed

during river floods, preventing the drainage of the flooded volume until the end of the event. As a result, these networks were not expected to contribute to the flood dynamics significantly, and hence they were neglected in the terrain description. Finally, it must be stressed that the database should be updated periodically to take into account possible significant changes to the landscape (i.e. construction/removal of relevant embankments) that are expected to affect the flood dynamic (Viero et al. (2019)).

Further considerations are required about the assumptions concerning the levee breach locations and dimensions for the simulated scenarios. First, focusing on the selection of the breach position, a distance of about 2 km between two consecutive sites was chosen. This pitch represents a compromise between the number of simulations to be performed (not so much for reducing the computational time, as for achieving a "manageable" number of scenarios for output analysis) and the possibility of capturing all the inundation patterns. In fact, while two close breaches often generate similar flooded areas, sometimes the

flooding evolution may change dramatically even for relatively close breaches. As an example, Figure 6 compares the inundated areas for two breach scenarios on the Secchia River, which are remarkably different: in the first case, the lowland area towards

the east is involved, whereas for the breach site immediately downstream, the flooding moves northwards due to the terrain morphology. This behaviour confirms that the levee breach locations should be carefully considered.

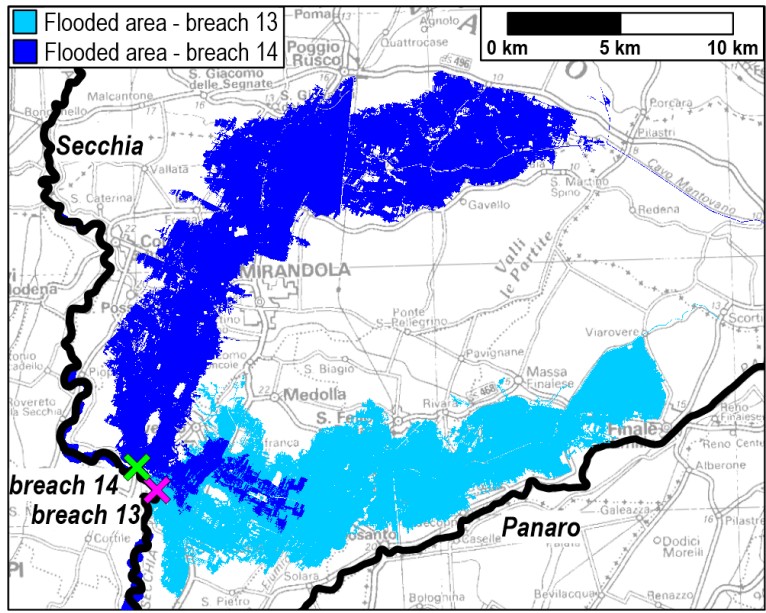

**Figure 6.** Example of the simulated flooded areas generated by two consecutive breaches (#13 and #14) on the Secchia River (the breach positions are indicated by crosses) for inflow B: the first breach scenario (cyan) mainly involves the eastern part of the domain, whereas the inundation for the second one (blue) moves northwards.

As regards the geometrical parameters assumed for the breach evolution, a sensitivity analysis on the breach final width and opening time was carried out, and the results for a given breach location on the Secchia River (see Fig. 3) are reported in Table 1 for inflows A and B. Maintaining the reference opening time $T$ (3 h for inflow A and 6 h for inflow B), additional tests were performed by varying the assumed final width $L$ (100 m) by $\pm30\%$ ($L = 70, 130$ m). As regards the opening time, $\pm50\%$ variations were explored (i.e. $T = 1.5, 4.5$ h for inflow A, and $T = 3, 9$ h for inflow B), considering a fixed value for the breach width ($L = 100$ m). The total volume flowing out of the breach and the extent reached by flooding at fixed times (24, 48 and 72 h after the breach opening) were used for comparing the different configurations. As expected, the results show that a larger breach, as well as a reduced opening time, slightly increases the flooded volume and area. However, the relative differences against the baseline simulation remain always below 10%: this confirms that the flooding scenarios are only marginally (less than linearly) influenced by the values assumed by these parameters.

Moreover, considering constant breach parameters ($L = 100$ m, $T = 3$ h), data reported in Table 1 also give an idea about the influence of the inflow condition on flooding results for this scenario: unsurprisingly, the total flooded volume for inflow B is 22% lower compared to inflow A, but the final flooded area is only 10% smaller. This means that breaching during a flood event with higher return period generates a more severe inundation on the lowland (i.e. higher water depths), while the affected area may be somewhat less influenced due to the terrain morphology and to the possible presence of obstacles that limit the flood

**Table 1.** Sensitivity analysis on the final breach width $L$ and opening time $T$ for one scenario on the Secchia River (see Fig. 3), with both inflows A and B. Results are compared based on the total outflow volume $vol$, and the flooded area 24, 48 and 72 h after the breach opening ($area_{24}$, $area_{48}$ and $area_{72}$, respectively). Their relative differences ($\Delta vol$, $\Delta area_{24}$, $\Delta area_{48}$, $\Delta area_{72}$) with reference to the baseline simulation are also reported for each tested configuration.

| Inflow | $L$ (m) | $T$ (h) | $vol$ ($10^6 m^3$) | $\Delta vol$ (%) | $area_{24}$ (km$^2$) | $\Delta area_{24}$ (%) | $area_{48}$ (km$^2$) | $\Delta area_{48}$ (%) | $area_{72}$ (km$^2$) | $\Delta area_{72}$ (%) |
|---|---|---|---|---|---|---|---|---|---|---|
| A | 100 | 3 | 52.75 | | 68.04 | | 99.08 | | 117.20 | |
| A | 70 | 3 | 48.35 | -8 | 63.52 | -7 | 93.89 | -5 | 111.99 | -4 |
| A | 130 | 3 | 55.09 | 4 | 70.11 | 3 | 101.55 | 2 | 120.29 | 3 |
| A | 100 | 1.5 | 54.31 | 3 | 69.53 | 2 | 100.55 | 1 | 119.05 | 2 |
| A | 100 | 4.5 | 51.19 | -3 | 65.97 | -3 | 97.59 | -2 | 115.39 | -2 |
| B | 100 | 6 | 38.24 | | 56.78 | | 84.07 | | 101.67 | |
| B | 70 | 6 | 35.20 | -8 | 51.71 | -9 | 80.00 | -5 | 96.47 | -5 |
| B | 130 | 6 | 39.89 | 4 | 58.99 | 4 | 86.54 | 3 | 104.71 | 3 |
| B | 100 | 3 | 41.00 | 7 | 61.02 | 7 | 87.94 | 5 | 105.83 | 4 |
| B | 100 | 9 | 35.51 | -7 | 50.92 | -10 | 80.52 | -4 | 97.40 | -4 |

propagation. This outcome is encouraging for the purpose of this work, because even for an actual flood event, whose inflow hydrograph can be quite different from the design hydrograph with assigned return period used for creating the database, at least the area possibly hit by flooding may be identified reasonably.

In this study, scenarios are not associated to their probability of occurrence, i.e. all breach locations are considered equally probable. This is consistent with the purpose of the methodology. However, if the same database of simulations had to be exploited for flood hazard (or even flood risk) assessment in the same area, information about the failure probability of the levee for each scenario would be required. This probability can be estimated by means of "fragility curves" for different levee failure mechanisms (Apel et al. (2006), Vorogushyn et al. (2010), Mazzoleni et al. (2013), Pinter et al. (2016)), sometimes called "levee failure functions", which depend both on the water level in the river and on the levee geometrical and geotechnical characteristics (often unknown). This analysis is beyond the scope of this paper, and is left to future developments.

## 5  Conclusions

With the aim of enhancing the resilience of lowland areas in case of levee breach occurrence, this paper defined a methodology for creating a database of hypothetical flood scenarios obtained from 2D numerical modelling, associated with different hydrological configurations and breach locations. The procedure, named RESILIENCE, was applied to a pilot area of about 1,100 km$^2$ in Northern Italy, but it can be extended to any other leveed river. The computational efficiency ensured by the adoption of the PARFLOOD parallel code allowed for the use of a high-resolution mesh (up to 2.5-5 m), while ratios of physical to computational time up to 80 were reached for some simulations. The application of numerical models to predict the flood dynamics provides useful data for emergency planning and management, and represents a fundamental tool for civil protection purposes and for increasing flood preparedness. Future developments of the methodology include: the expansion of the current database for the pilot area (e.g. other hydrological inputs, breaching along the Po River, multiple breach openings), the identification of the most probable failure locations, and the application of the RESILIENCE procedure to other rivers and lowland areas.

Finally, support and assistance will be provided to public administrations for the correct interpretation and employment of the simulation results during civil protection planning.

*Video supplement.* A simulation visualization of a hypothetical levee breach-induced flooding on the Secchia River is available as additional material at https://doi.org/10.5446/44534

*Author contributions.* PM and RV designed the RESILIENCE project and identified the pilot area. RV, SD, AF, PM contributed to the
425 implementation of the PARFLOOD 2D-SWE code. AF initially calibrated the Secchia River model, while SD calibrated the Panaro River model and carried out all the levee breach simulations. AF and SD wrote the paper in consultation with RV and PM. All authors reviewed the final manuscript.

*Competing interests.* The authors declare that they have no conflict of interest.

*Acknowledgements.* This work was financially supported by the Territorial Safety and Civil Protection Agency of Emilia-Romagna Region
(Agenzia per la Sicurezza Territoriale e la Protezione Civile della Regione Emilia Romagna). The Interregional Agency for the Po River (Agenzia Interregionale per il fiume Po) and the Po River Basin Authority (Autorità di Bacino distrettuale del fiume Po) are gratefully acknowledged for providing data for the Secchia and Panaro Rivers. The publication of this paper is financially supported by the Center for Studies in European and International Affairs (CSEIA) of the University of Parma under the OPEN-UP (Outgoing Publications, Essays and Networks) call. This research benefits from the HPC (High Performance Computing) facility of the University of Parma. This work was
partially supported by Ministry of Education, Universities and Research under the Scientific Independence of young Researchers project, grant number RBSI14R1GP, CUP code D92I15000190001. The authors are grateful to the editor and the reviewers for their constructive comments on the early version of this manuscript.

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
