# Peer review of "Enhancing the resilience to flooding induced by levee breaches in lowland areas: a methodology based on numerical modelling"

_Natural Hazards and Earth System Sciences, 2019_

## Short Comment (SC1) · 2 Jun 2019

The study proposed a methodology to create a database of numerically simulated flooding scenarios with embankment failures with the objective of improving resilience to flooding and increasing hazard preparedness in lowlands with levee breach-induced inundations.

The study is well worth investigating and the paper was well written. The application of the proposed model may have significant implications for hazard preparedness. Here are my concerns:

[Figure]

Although the authors briefly mentioned that optimization techniques have been implemented (see lines 29 and 30 of the manuscript) they have failed to cite recent relevant studies that have applied mathematical modeling and optimization strategies. It is important that the authors emphasize this more in this study. The following are some critical references:

1) Dulebenets, M. A., Pasha, J., Abioye, O. F., Kavoosi, M., Ozguven, E. E., Moses, R., Boot, W., Sando, T. (2019). Exact and heuristic solution algorithms for efficient emergency evacuation in areas with vulnerable populations. International Journal of Disaster Risk Reduction.

2) Trivedi, A., Singh, A.. (2017). A hybrid multi-objective decision model for emergency shelter location-relocation projects using fuzzy analytic hierarchy process and goal programming approach, Int. J. Proj. Manag., 35 (5), pp. 827-840

3) Pel, A., Bliemer, M., and Hoogendoorn, S. (2012). A review on travel behaviour modelling in dynamic traffic simulation models for evacuations, Transportation, 39, pp. 97-123.

---

## Author Comment (AC1) · 5 Jun 2019

The authors gratefully acknowledge the positive and constructive comment of Dr. Olumide Abioye.

The recent studies mentioned in the comment, concerning optimization strategies applied to emergency evacuation, will be considered in the revised version of the manuscript.

---

## Referee Comment (RC1) · Anonymous Referee #1 · 20 Jun 2019

I enjoy reading the paper by Ferrari and co-authors. It presents a method aimed at improving the resilience of lowland areas that are subject to flooding caused by possible levee failures. The method is simple, as it consists of composing a database of flooding dynamics caused by a (large) number of simulated levee breaches a priori, which allows knowing with good accuracy the flooding dynamics in case of a real levee failure by choosing the simulated event that is most similar in terms of breach locations (and possibly flood magnitude).

I can confirm from my experience that such an information could be extremely useful for civil protection purposes. I am thinking of the case of a town that ten years ago

was completely flooded about a day and a half after the occurrence of a levee failure, without undertaking significant countermeasures due to the lack of knowledge about flooding propagation in that area.

The text is clear and generally well written. The English could be slightly improved in term of readability with a careful proofreading.

It could be worth adding some discussion on the role of the drainage network that typically dissects rural anthropogenic lowlands. Drainage networks, which comprise ditches and channels of gradually increasing size, as well as small obstructions, were proven to affect the flow dynamics at a local scale, encompassing flood formation, the speed of the submerging wave and the flow direction (Hailemariam et al., 2014; Viero et al., 2014; Viero and Valipour, 2017). On the other hand, it must be recognized that the present study deals with the simulation of major flood events, as those caused by levee breaches generally are, and it is reasonable to assume that relatively small landscape features produce negligible effects in such cases.

Finally, I suggest stressing that such a database should be updated when significant modifications affect the landscape and, particularly, the topography of the floodable area, particularly for embankment construction and/or removal, as they can change the flood dynamics dramatically and, often, in unexpected fashions (e.g., Viero et al,, 2019).

ADDITIONAL REFERENCES

Hailemariam, F.M., Brandimarte, L., Dottori, F., 2014. Investigating the influence of minor hydraulic structures on modeling flood events in lowland areas. Hydrol. Process. 28, 1742–1755. doi:10.1002/hyp.9717

Viero, D.P., Peruzzo, P., Carniello, L., Defina, A., 2014. Integrated mathematical modeling of hydrological and hydrodynamic response to rainfall events in rural lowland catchments. Water Resour. Res. 50, 5941–5957. doi:10.1002/2013WR014293

Viero, D.P., Valipour, M., 2017. Modeling anisotropy in free-surface overland and shallow inundation flows. Adv. Water Resour. 104, 1–14. doi:10.1016/j.advwatres.2017.03.007

Viero, D.P., Roder, G., Matticchio, B., Defina, A., Tarolli, P., 2019. Floods, landscape modifications and population dynamics in anthropogenic coastal lowlands: The Polesine (northern Italy) case study. Sci. Total Environ. 651, 1435–1450. doi:10.1016/j.scitotenv.2018.09.121
* * *

---

## Referee Comment (RC2) · Anonymous Referee #2 · 11 Sep 2019

The paper presents an interesting contribution to the journal, offering a novel approach to improving resilience to flooding and increasing preparedness to face levee breach-induced inundations. however, i have some major concerns related to the current version of the manuscript. Overall, some parts lack clarity, and this might cause some confusion while reading the article. Also, English needs polishing before the paper is ready for publication.

As the paper deals with lowlands and human modifications (levees), it should be important to include some contextual works on the role of levees and embankments to contributing to flooding (i.e. Black 2008; Munoz et al., 2018) and on the importance

of the artificial drainage network and landscape changes in contributing to floods (i.e. Wohl 2019a,b; Pijl, Brauer, Sofia, Teuling, & Tarolli, 2018; Sofia et al., 2019). The discussion of the results should also be framed into this wider context. Currently, it is much focussed on the technical domain (computer requirements, time for simulation etc), but the paper would benefit a wider audience if the results were framed into the larger picture of lowlands and flood risk.

A greater concern emerges for the paper structure. In my opinion, the paper structure is very confused, and the chapters are currently disorganized proposing a mixture of literature review, method description, and results altogether. There are a lot of references to what should or should not be done, according to a literature review, rather than a focus on the novelty of the proposed approach, and this makes the text hard to follow. The paper should at first describe what the RESILIENCE project is (beginning of chapter 3) and then describe the methodology proposed in this paper (i.e. ParFlood and why it is novel/Accurate), and further proceed to describe the setting for the current simulation. Currently, much of the description is about previous works and all possible approaches, but this 'distract' from the description of the actual method proposed. The authors should consider rewording the text, so that it is clear what are the novelties and strengths of this work, as compared to past ones.

Within the methods, also, a lot of parameters are case-specific and it is not clear how they should be 'tuned' for further application of this approach in different study areas. For example, it is not clear what the 'hydrological scenario' are. Do they come from simulated flows? do they come from actual data? if they come from simulated flows, how are these accomplished? Also, the choice of the return period for inflows A and B is not clear. was this return period previously analyzed and identified? how? [this latter confusion probably emerge from some lack of clarity in the manuscript]. Should the parameters be optimized for future studies, if so how?

A further issue is that the authors state that 'Compared to previous studies on flooding induced by levee breaches, the proposed methodology benefits from the adoptions of an accurate and fast numerical model and of high-resolution meshes', but the manuscript does not present any actual comparison with previous studies, but it only showcases a literature review on them.

I believe addressing these issues would add value to the paper, and would make this work useful to a wider scientific audience.

References Black, H., 2008. Unnatural disaster: human factors in the Mississippi floods. Environmental health perspectives, 116(9), pp.A390–3. Available at: http://dx.doi.org/10.1289/ehp.116-a390. Munoz, S. E., Giosan, L., Therrell, M. D., Remo, J. W. F., Shen, Z., Sullivan, R. M., . . . Donnelly, J. P. (2018). Climatic control of Mississippi River flood hazard amplified by river engineering. Nature, 556(7699), 95–98. https://doi.org/10.1038/nature26145 Pijl, A., Brauer, C. C., Sofia, G., Teuling, A. J., & Tarolli, P. (2018). Hydrologic impacts of changing land use and climate in the Veneto lowlands of Italy. Anthropocene, 22. https://doi.org/10.1016/j.ancene.2018.04.001 Sofia, G., Ragazzi, F., Giandon, P., Dalla Fontana, G., Tarolli, P., Fontana, G. D., & Tarolli, P. (2019). On the linkage between runoff generation, land drainage, soil properties, and temporal patterns of precipitation in agricultural floodplains. Advances in Water Resources, 124, 120–138. https://doi.org/10.1016/j.advwatres.2018.12.003 Wohl, E., 2019a. Forgotten Legacies: Understanding and Mitigating Historical Human Alterations of River Corridors. Water Resources Research. Available at: http://dx.doi.org/10.1029/2018wr024433. Wohl, E., 2019b. Forgotten Legacies: Understanding Human Influences on Rivers. Eos, 100. Available at: http://dx.doi.org/10.1029/2019eo127853.

---

## Author Comment (AC3) · 23 Sep 2019

**The authors gratefully acknowledge the positive and constructive review of the anonymous Referee. In this document the comments provided by the Referee are reported in italic, whereas the authors' response and indications about the original paper modifications are marked in bold fonts.**

*The paper presents an interesting contribution to the journal, offering a novel approach to improving resilience to flooding and increasing preparedness to face levee breach-induced inundations. However, I have some major concerns related to the current*

*version of the manuscript. Overall, some parts lack clarity, and this might cause some confusion while reading the article. Also, English needs polishing before the paper is ready for publication.*

**We thank the Reviewer for his comment. The entire manuscript will be carefully revised.**

*As the paper deals with lowlands and human modifications (levees), it should be important to include some contextual works on the role of levees and embankments to contributing to flooding (i.e. Black 2008; Munoz et al., 2018) and on the importance of the artificial drainage network and landscape changes in contributing to floods (i.e. Wohl 2019a,b; Pijl, Brauer, Sofia, Teuling, & Tarolli, 2018; Sofia et al., 2019). The discussion of the results should also be framed into this wider context. Currently, it is much focussed on the technical domain (computer requirements, time for simulation etc), but the paper would benefit a wider audience if the results were framed into the larger picture of lowlands and flood risk.*

**We agree that flooding events represent a crucial task for different research branches. Therefore, in the revised manuscript, we will reword both the introduction and the discussion section in order to better discuss the role exerted by levees on the flooding of lowlands. Particularly, we will deepen the description of the levee-effect problem and we will add references to the mentioned contributions.**
**As regards the drainage network, we will add some discussion about the fact that the drainage networks only influence the flood dynamics at a local scale, for example by defining preferential pathways. However, in the scenarios here considered, the flood volume (in the order of $10^7$ m$^3$) largely exceeds the discharge capacity of the drainage systems. It is also relevant to notice that**

**most of the minor channels are equipped with gates that are kept closed at the passage of huge flood waves in the river, and hence they do not contribute to the drainage of the flooded volume until the end of the event. As a result, these networks are not expected to significantly contribute to the flood dynamics, and hence they were neglected in the terrain description, also to avoid the excessive increase in the number of computational cells.**

*A greater concern emerges for the paper structure. In my opinion, the paper structure is very confused, and the chapters are currently disorganized proposing a mixture of literature review, method description, and results altogether. There are a lot of references to what should or should not be done, according to a literature review, rather than a focus on the novelty of the proposed approach, and this makes the text hard to follow. The paper should at first describe what the RESILIENCE project is (beginning of chapter 3) and then describe the methodology proposed in this paper (i.e. ParFlood and why it is novel/Accurate), and further proceed to describe the setting for the current simulation. Currently, much of the description is about previous works and all possible approaches, but this 'distract' from the description of the actual method proposed. The authors should consider rewording the text, so that it is clear what are the novelties and strengths of this work, as compared to past ones.*

**We appreciate this suggestion and accordingly we will modify the structure of the manuscript in order to clearly distinguish among literature review, the presentation of the RESILIENCE project, and the results obtained by applying the proposed methodology to the study area, and to highlight the novelties of the approach.**
**In particular, we will start by presenting the RESILIENCE project and the methodology in general, and then we will describe the application to the case study, which will also allow us to further discuss the strengths of this work.**

*Within the methods, also, a lot of parameters are case-specific and it is not clear how they should be 'tuned' for further application of this approach in different study areas. For example, it is not clear what the 'hydrological scenario' are. Do they come from simulated flows? do they come from actual data? if they come from simulated flows, how are these accomplished? Also, the choice of the return period for inflows A and B is not clear. Was this return period previously analyzed and identified? how? [this latter confusion probably emerge from some lack of clarity in the manuscript]. Should the parameters be optimized for future studies, if so how?*

**We thank the Referee for this useful comment that allows us to clarify some further aspects of the proposed methodology. In the revised paper we will better stress that the methodology is general and that it can be applied to any leveed river. Therefore, we will explain that the values of the breach parameters adopted for the pilot area (i.e. location point, width, evolution time) are provided as examples, and this does not prevent the extension of the RESILIENCE project to different areas.**
**In this context, we will also clarify the role of the hydrological scenarios (A and B in Sect. 3.2) in the creation of the database of flooding scenarios. In short, the discharge hydrographs adopted as upstream boundary condition are simply synthetic design hydrographs with assigned return period, derived from previous hydrological studies, which in our case were provided by the Po River Basin Authority. Please notice that for most rivers such hydrographs are already available, often employed for creating flood hazard/risk maps according to the EU Floods Directive. We performed preliminary simulations of the propagation of floods with different return periods (e.g. 20, 50, 100, 200 years), and identified the one corresponding to incipient overtopping: this was inflow A. For example, inflow A was the 50 years-hydrograph for the Secchia River, and the 100 years-**

**one for the Panaro River. Then, a lower return period hydrograph (20 years) was chosen in order to consider levee collapses due to piping or other mechanisms, during a flood event that does not induce overtopping: this was inflow B.**

*A further issue is that the authors state that 'Compared to previous studies on flooding induced by levee breaches, the proposed methodology benefits from the adoptions of an accurate and fast numerical model and of high-resolution meshes', but the manuscript does not present any actual comparison with previous studies, but it only showcases a literature review on them.*

**The Referee is right. The paper does not compare the results of the RESILIENCE methodology with those of previous studies, and the sentence "compared to previous studies .." was adopted to refer to literature studies. We agree with the Reviewer that this can lead to misunderstanding, hence we will correct this sentence in the revised paper accordingly.**

*I believe addressing these issues would add value to the paper, and would make this work useful to a wider scientific audience.*

**The authors wish to thank the anonymous Referee for his positive overview about the manuscript.**

*References Black, H., 2008. Unnatural disaster: human factors in the Mississippi floods. Environmental health perspectives, 116(9), pp.A390–3. Available at: http://dx.doi.org/10.1289/ehp.116-a390.*
*Munoz, S. E., Giosan, L., Therrell, M.D., Remo, J. W. F., Shen, Z., Sullivan, R. M., : : : Donnelly, J. P. (2018). Climatic control of Mississippi River flood hazard amplified by*

*river engineering. Nature, 556(7699), 95–98. https://doi.org/10.1038/nature26145*

*Pijl, A., Brauer, C.C., Sofia, G., Teuling, A. J., & Tarolli, P. (2018). Hydrologic impacts of changing land use and climate in the Veneto lowlands of Italy. Anthropocene, 22. https://doi.org/10.1016/j.ancene.2018.04.001*

*Sofia, G., Ragazzi, F., Giandon, P., Dalla Fontana, G., Tarolli, P., Fontana, G. D., & Tarolli, P. (2019). On the linkage between runoff generation, land drainage, soil properties, and temporal patterns of precipitation in agricultural floodplains. Advances in Water Resources, 124, 120–138. https://doi.org/10.1016/j.advwatres.2018.12.003*

*Wohl, E., 2019a. Forgotten Legacies: Understanding and Mitigating Historical Human Alterations of River Corridors. Water Resources Research. Available at: http://dx.doi.org/10.1029/2018wr024433.*

*Wohl, E., 2019b. Forgotten Legacies: Understanding Human Influences on Rivers. Eos, 100. Available at: http://dx.doi.org/10.1029/2019eo127853.*

---

## Author Response (AR1)

Prof. Paolo Tarolli,

Editor of Natural Hazards and Earth System Sciences

09/10/2019

**Re: Manuscript nhess-2019-132**

Thank you for your comments regarding our manuscript entitled "A methodology based on numerical models for enhancing the resilience to flooding induced by levee breaches in lowland areas" by A. Ferrari, S. Dazzi, R. Vacondio, and P. Mignosa. We have now updated the manuscript following the suggestions of the Reviewers after their analysis of the paper and the comment of Dr. Olumide Abioye. Moreover, according to your suggestions, we have provided a colour legend in Figure 6, and we have carefully revised the manuscript deserving attention to the English language.

Please find attached to this letter a description of changes and our replies to each comment along with an updated manuscript.

We hope that all the points raised by the Reviewers have been satisfactorily addressed. We wish to kindly thank the referees for their careful reviews and invaluable comments and hope to hear from you again on the status of the manuscript.

Yours sincerely,

Alessia Ferrari
PhD, Research Assistant
Department of Engineering and Architecture
University of Parma

**RESPONSE TO REFEREE #1:**

**The authors gratefully acknowledge the positive and constructive review of the anonymous Referee. In this document the comments provided by the Referee are reported in italic, whereas the authors' response and indications about the original paper modifications are marked in bold fonts.**

*I enjoy reading the paper by Ferrari and co-authors. It presents a method aimed at improving the resilience of lowland areas that are subject to flooding caused by possible levee failures. The method is simple, as it consists of composing a database of flooding dynamics caused by a (large) number of simulated levee breaches a priori, which allows knowing with good accuracy the flooding dynamics in case of a real levee failure by choosing the simulated event that is most similar in terms of breach locations (and possibly flood magnitude). I can confirm from my experience that such an information could be extremely useful for civil protection purposes. I am thinking of the case of a town that ten years ago was completely flooded about a day and a half after the occurrence of a levee failure, without undertaking significant countermeasures due to the lack of knowledge about flooding propagation in that area.*

**The authors wish to thank the anonymous Referee for his positive overview about the manuscript.**

*The text is clear and generally well written. The English could be slightly improved in term of readability with a careful proofreading.*

**We thank the Reviewer for his suggestion. The entire manuscript has been carefully revised.**

*It could be worth adding some discussion on the role of the drainage network that typically dissects rural anthropogenic lowlands. Drainage networks, which comprise ditches and channels of gradually increasing size, as well as small obstructions, were proven to affect the flow dynamics at a local scale, encompassing flood formation, the speed of the submerging wave and the flow direction (Hailemariam et al., 2014; Viero et al., 2014; Viero and Valipour, 2017). On the other hand, it must be recognized that the present study deals with the simulation of major flood events, as those caused by levee breaches generally are, and it is reasonable to assume that relatively small landscape features produce negligible effects in such cases.*

**We thank the Referee for this comment and we agree that drainage networks and microtopography (i.e. tillage feature, ditches) influence the flood dynamics at a local scale, for example by defining preferential pathways. However, as already pointed out by the Referee, the maximum discharge through the breaches here considered (in the order of $10^2 \, \text{m}^3/\text{s}$) largely exceeds the discharge capacity of the drainage systems. It is also relevant to notice that most of the minor channels are equipped with gates that are kept closed at the passage of huge flood waves in the river, and hence they do not contribute to the drainage of the flooded volume until the end of the event. Even the possible presence of pumping stations**

is not relevant during the event, considering the extension of the inundations modelled here. As a result, these networks are not expected to significantly contribute to the flood dynamics, and hence they were neglected in the terrain description to avoid the excessive increase in the number of computational cells. In fact, the riverbed, the levees, the main artificial embankments and channels were described with the maximum resolution (5 m), whereas the description of the remaining lowlands with small channels would have led to an 80% increase in the number of cells. In addition to this, most of these micro-features would have required an even finer resolution (e.g. 1 m), which would have further increased the computational time.

In the revised paper, we have clarified the reasons for neglecting the drainage network by adding the following text (Sect. 4, line 33 (page 12) - line 5 (page 13)):

"Still focusing on topographic information and spatial resolution it is relevant to notice that drainage networks with relatively narrow channels were not described in detail in the computational grid. In fact, even though microtopography (i.e. tillage feature, ditches) determines preferential pathways for very shallow flows (Viero and Valipur, 2017; Hailemariam et al., 2014), the maximum discharge through the breaches here considered (in the order of $10^2$ m$^3$/s) largely exceeds the discharge capacity of the drainage systems. Moreover, most of the minor channels are equipped with sluice gates that are kept closed during river floods, preventing the drainage of the flooded volume until the end of the event. As a result, these networks were not expected to contribute to the flood dynamics significantly, and hence they were neglected in the terrain description."

*Finally, I suggest stressing that such a database should be updated when significant modifications affect the landscape and, particularly, the topography of the floodable area, particularly for embankment construction and/or removal, as they can change the flood dynamics dramatically and, often, in unexpected fashions (e.g., Viero et al., 2019).*

We thank the Referee for this comment. Therefore, in the discussion section (Sect. 4, lines 6-8 (page 13)), we have added the following sentence:

"Finally, it must be stressed that the database should be updated periodically to take into account possible significant changes to the landscape (i.e. construction/removal of relevant embankments) that are expected to affect the flood dynamic (Viero et al., 2019)."

*ADDITIONAL REFERENCES*

*Hailemariam, F.M., Brandimarte, L., Dottori, F., 2014. Investigating the influence of minor hydraulic structures on modeling flood events in lowland areas. Hydrol. Process. 28, 1742–1755. doi:10.1002/hyp.9717*

Viero, D.P., Peruzzo, P., Carniello, L., Defina, A., 2014. Integrated mathematical modeling of hydrological and hydrodynamic response to rainfall events in rural lowland catchments. Water Resour. Res. 50, 5941–5957. doi:10.1002/2013WR014293

Viero, D.P., Valipour, M., 2017. Modeling anisotropy in free-surface overland and shallow inundation flows. Adv. Water Resour. 104, 1–14. doi:10.1016/j.advwatres.2017.03.007

Viero, D.P., Roder, G., Matticchio, B., Defina, A., Tarolli, P., 2019. Floods, landscape modifications and population dynamics in anthropogenic coastal lowlands: The Polesine (northern Italy) case study. Sci. Total Environ. 651, 1435–1450. doi:10.1016/j.scitotenv.2018.09.121

**RESPONSE TO REFEREE #2:**

**The authors gratefully acknowledge the positive and constructive review of the anonymous Referee. In this document the comments provided by the Referee are reported in italic, whereas the authors' response and indications about the original paper modifications are marked in bold fonts**.

*The paper presents an interesting contribution to the journal, offering a novel approach to improving resilience to flooding and increasing preparedness to face levee breach-induced inundations. However, I have some major concerns related to the current version of the manuscript. Overall, some parts lack clarity, and this might cause some confusion while reading the article. Also, English needs polishing before the paper is ready for publication.*

**We thank the Reviewer for his comment. The entire manuscript has been carefully revised.**

*As the paper deals with lowlands and human modifications (levees), it should be important to include some contextual works on the role of levees and embankments to contributing to flooding (i.e. Black 2008; Munoz et al., 2018) and on the importance of the artificial drainage network and landscape changes in contributing to floods (i.e. Wohl 2019a,b; Pijl, Brauer, Sofia, Teuling, & Tarolli, 2018; Sofia et al., 2019). The discussion of the results should also be framed into this wider context. Currently, it is much focussed on the technical domain (computer requirements, time for simulation etc), but the paper would benefit a wider audience if the results were framed into the larger picture of lowlands and flood risk.*

**We agree that flooding events represent a crucial task for different research branches. Therefore, we have reworded the text in Section 1 (lines 22-27) to better explain the "levee-effect" problem, which actually represents one of the main reason to study levee breach-induced flooding, as follows:**

**"Among the possible causes of flooding, levee breaching deserves special attention. Due to the well-known "levee-effect" phenomenon, structural flood protection systems, such as levees, determine an increase in flood exposure. In fact, the presence of this hydraulic defence creates a feeling of safety among people living in flood-prone areas, both resulting in the growing of settlements, and in the reduction of preparedness, hence in the increase of vulnerability in those areas (Di Baldassarre et al., 2015). As a result, more people are exposed to less frequent but more devastating floods, for which the statistical frequency is difficult to assess, due to the historical changes in river systems (Black et al., 2008)."**

**Moreover, in the discussion section (Sect. 4, line 28 (page 11) - line 3 (page 12)) we have added the following sentence to briefly recall the ecological and socio-economic consequences related to the building of structural protection systems:**

"Due to climate change and population growth, structural protection systems such as levees have been often adopted to increase protection against floods. However, this kind of defence presents several ecological (e.g. hydraulic decoupling between the river and its floodplain, loss of biodiversity, change in groundwater levels, increase in greenhouse gas emission) and socio-economic consequences (Auerswald et al., 2019).

Focusing on the "levee-effect" phenomenon and on the issue of adequately considering the residual flood risk related to levees, the main goal of this paper was to define a methodology based on the use of numerical models to enhance the resilience of lowland areas in case of levee breach occurrence."

As regards the drainage network, we have added some discussion about the fact that the drainage networks only influence the flood dynamics at a local scale, for example by defining preferential pathways. However, the maximum discharge through the breaches here considered (in the order of $10^2$ $m^3$/s) largely exceeds the discharge capacity of the drainage systems. Finally, it is also relevant to notice that most of the minor channels are equipped with gates that are kept closed at the passage of huge flood waves in the river, and hence they do not contribute to the drainage of the flooded volume until the end of the event. As a result, these networks are not expected to significantly contribute to the flood dynamics, and hence they were neglected in the terrain description to avoid the excessive increase in the number of computational cells.

In the revised paper, we have clarified the reasons for neglecting the drainage network by adding the following text (Sect. 4, line 33 (page 12) - line 5 (page 13)):

"Still focusing on topographic information and spatial resolution it is relevant to notice that drainage networks with relatively narrow channels were not described in detail in the computational grid. In fact, even though microtopography (i.e. tillage feature, ditches) determines preferential pathways for very shallow flows (Viero and Valipur, 2017; Hailemariam et al., 2014), the maximum discharge through the breaches here considered (in the order of $10^2$ $m^3$/s) largely exceeds the discharge capacity of the drainage systems. Moreover, most of the minor channels are equipped with sluice gates that are kept closed during river floods, preventing the drainage of the flooded volume until the end of the event. As a result, these networks were not expected to contribute to the flood dynamics significantly, and hence they were neglected in the terrain description."

Finally, in order to better stress that significant landscape changes, which can impact on the flood dynamics, have to be taken into account once the database of simulations is created, we have added the following sentence in the discussion section (Sect. 4, lines 6-8 (page 13)):

**"Finally, it must be stressed that the database should be updated periodically to take into account possible significant changes to the landscape (i.e. construction/removal of relevant embankments) that are expected to affect the flood dynamic (Viero et al., 2019)."**

*A greater concern emerges for the paper structure. In my opinion, the paper structure is very confused, and the chapters are currently disorganized proposing a mixture of literature review, method description, and results altogether. There are a lot of references to what should or should not be done, according to a literature review, rather than a focus on the novelty of the proposed approach, and this makes the text hard to follow. The paper should at first describe what the RESILIENCE project is (beginning of chapter 3) and then describe the methodology proposed in this paper (i.e. ParFlood and why it is novel/Accurate), and further proceed to describe the setting for the current simulation. Currently, much of the description is about previous works and all possible approaches, but this 'distract' from the description of the actual method proposed. The authors should consider rewording the text, so that it is clear what are the novelties and strengths of this work, as compared to past ones.*

**We appreciate this suggestion and accordingly we have changed the structure of the paper in order to better mark the novelties of the work. Particularly, we reworded the text in order to distinguish among the literature review, the presentation of the RESILIENCE project, and its application to the study area. Therefore, in Sect. 2, the novel methodology is presented, and general guidelines concerning choice of the numerical model, topographic data, hydrological scenarios, levee breaches locations, and outputs are discussed.**

**Following the Reviewer suggestion, in the "numerical model" subsection (Sect. 2.1) we have initially pointed out the strength and weakness of some current numerical models used to model free surface flows, particularly levee breach-induced flooding, and then described the advantages of the PARFLOOD model here used.**

**Moreover, in order to stress that the proposed methodology is general and that it can be potentially applied to any leveed river, general guidelines concerning the RESILIENCE project are provided in Sect. 2, whereas the application to a pilot area in Northern Italy (e.g. description of the input parameters, analysis of the results) is presented in Sect. 3.**

*Within the methods, also, a lot of parameters are case-specific and it is not clear how they should be 'tuned' for further application of this approach in different study areas. For example, it is not clear what the 'hydrological scenario' are. Do they come from simulated flows? do they come from actual data? if they come from simulated flows, how are these accomplished? Also, the choice of the return period for inflows A and B is not clear. Was this return period previously analyzed and identified? how? [this latter confusion probably*

*emerge from some lack of clarity in the manuscript]. Should the parameters be optimized for future studies, if so how?*

We thank the Referee for this useful comment that allows us to clarify some further aspects of the proposed methodology. In the revised paper, also by separating the sections concerning the description of the methodology (Sect. 2) and its application to the chosen pilot area (Sect. 3), we have better stressed that the methodology is general and that it can be applied to any leveed river.

Moreover, we have reworded Sect. 2.3 in order to clarify the role of the hydrological scenarios in the creation of the database of flooding scenarios, as follows:

"Discharge hydrographs with a specific return period are assigned as upstream boundary condition. Sometimes these hydrographs are already available from previous hydrological studies, and can be provided by local River Basin Authorities; otherwise, they can be derived from rainfall-runoff modelling or from statistical analyses of recorded discharge hydrographs (e.g. Tomirotti and Mignosa, 2017).

For the purpose of this study, multiple hydrological conditions should be considered, in order to cover possible configurations characterized by different breach triggering mechanisms, flood volumes, etc. At least two different discharge hydrographs should be considered for each breach location, even though the simulation database can be extended with more hydrological inputs if needed. The first case ("inflow A") corresponds to the condition for which the water surface elevation reaches the levee crest somewhere along the river, thus generating overtopping. The second configuration ("inflow B") concerns a flood event with a lower return period, for which the levee is never overtopped, but other mechanisms might induce the levee collapse. In fact, earthen levees can also experience breaching for piping and internal erosion processes, even when water levels remain below the levee crest. Besides, the dens of burrowing animals (e.g. porcupine, badger, nutria) were recently identified as another possible cause for breach triggering (Viero et al., 2013; Orlandini et al., 2015). Incidentally, the collapse of an embankment during a flood event with a relatively low return period can be very threatening for human lives because the highest warning thresholds may not be reached, and population can be unprepared to face flooding.

The choice of the return period for inflows A and B is river-dependent, because it is influenced by the design return period of the levee system, by the presence of other flood control structures, etc. In general, preliminary simulations of the propagation of flood waves with different return periods (e.g. 10, 20, 50, 100, 200, 500 years) for each river should be performed, and the event that corresponds to incipient overtopping can be identified as inflow A. Then, a higher frequency hydrograph can be selected as inflow B, in order to consider levee collapses due to piping or other mechanisms during a flood event that does not induce overtopping (for example, when a specific freeboard is guaranteed).

The discharge hydrographs thus obtained are imposed as upstream boundary condition for the levee breach scenarios."

*A further issue is that the authors state that 'Compared to previous studies on flooding induced by levee breaches, the proposed methodology benefits from the adoptions of an accurate and fast numerical model and of high-resolution meshes', but the manuscript does not present any actual comparison with previous studies, but it only showcases a literature review on them.*

**The Referee is right. The paper does not compare the results of the RESILIENCE methodology with those of previous studies, and the sentence "compared to previous studies ..", which in the original manuscript was adopted to refer to the literature studies, has been removed in the revised paper.**

*I believe addressing these issues would add value to the paper, and would make this work useful to a wider scientific audience.*

**The authors wish to thank the anonymous Referee for his positive overview about the manuscript.**

**RESPONSE TO SHORT COMMENT: Dr. Olumide Abioye**

**The authors gratefully acknowledge the positive comment of Dr. Olumide Abioye. In this document the comment provided by Dr. Olumide Abioye is reported in italic, whereas the authors' response and indications about the original paper modifications are marked in bold fonts**.

*The study proposed a methodology to create a database of numerically simulated flooding scenarios with embankment failures with the objective of improving resilience to flooding and increasing hazard preparedness in lowlands with levee breach-induced inundations.*

*The study is well worth investigating and the paper was well written. The application of the proposed model may have significant implications for hazard preparedness. Here are my concerns:*

*Although the authors briefly mentioned that optimization techniques have been implemented (see lines 29 and 30 of the manuscript) they have failed to cite recent relevant studies that have applied mathematical modeling and optimization strategies. It is important that the authors emphasize this more in this study. The following are some critical references:*

*1) Dulebenets, M. A., Pasha, J., Abioye, O. F., Kavoosi, M., Ozguven, E. E., Moses, R., Boot, W., Sando, T. (2019). Exact and heuristic solution algorithms for efficient emergency evacuation in areas with vulnerable populations. International Journal of Disaster Risk Reduction.*

*2) Trivedi, A., Singh, A.. (2017). A hybrid multi-objective decision model for emergency shelter location-relocation projects using fuzzy analytic hierarchy process and goal programming approach, Int. J. Proj. Manag., 35 (5), pp. 827-840*

*3) Pel, A., Bliemer, M., and Hoogendoorn, S. (2012). A review on travel behaviour modelling in dynamic traffic simulation models for evacuations, Transportation, 39, pp. 97-123.*

**We thank Dr. Olumide Abioye for his comment. In Sect. 4 (page 12, lines 10-14) we have added the following text:**

**"Besides these emergencies, the results of each simulation, also combined for global considerations, allow for an improvement of evacuation and defence system planning. In this context, advanced optimization-based tools and algorithms (Dulebenets et al., 2019a,b) can be exploited to create emergency evacuation plans that efficiently minimize individuals travel time during a natural hazard."**

The comparison between the original paper and the revised one is shown in the following pages. Since the manuscript has been completely reworded according to the Reviewer' suggestions, several parts have been moved or changed. As a result, the track-changed manuscript is not clear, and we would kindly suggest to refer to the revised manuscript.

[revised manuscript text omitted]
 hydrological events, characterized by different frequency, are considered. 2D numerical simulations of the flood scenarios resulting from each combination of breach position and upstream boundary 25   condition are performed, in order to create a large database of simulations covering any potential real levee breach event in that area in the best possible way. The results of these simulations, made available to public administrations, can be fundamental not only for emergency planning, but also for civil protection immediate action during actual flood events.

**The 3 Application of the RESILIENCE project to a pilot area for this in Northern Italy**

In this section, an example of application of the proposed methodology is presented. The pilot area for the RESILIENCE 30   project (Figure 1) is at the boundary of two regions (Emilia-Romagna and Lombardia,), in Northern Italy). This territorial

unit is about 1,100 km² wide, and is delimited by the Po River (North) and by its two right tributaries Secchia (West) and Panaro (East). The city of Modena bounds the area to the South. This lowland area can be potentially affected by flooding events caused by breaches from the 83 km-long right levee of the Secchia River  and/or from the 67 km-long left levee of the Panaro River .

5 This study area was selected for several reasons. First, the latest report of the Italian Institute for Environmental Protection and Research (ISPRA, 2018) showed that Emilia-Romagna is the Italian region with the highest level of exposed population (up to 2.7 million exposed inhabitants out of 4.3), buildings and areas for both high (return period of 20-50 years) and medium (return period of 100-200 years) flood frequency. Moreover, the middle-lower basin of the Po River was subject to levee breach-induced floods several times in the last 150 years, either from the main river levees or

10 from its leveed tributaries (e.g. Di Baldassarre et al., 2009; Masoero et al., 2013; D'Oria et al., 2015; Dazzi et al., 2019), often with devastating consequences. Finally, the Secchia and Panaro rivers experienced levee breach events in the past, even without overtopping and during the occurrence of floods with low/medium return periods. In particular, the most recent event that occurred in this area was the flood originated by a bank failure on the Secchia River in 2014 (Vacondio et al., 2016), which caused roughly 500 million euros losses. This event raised awareness of the huge damages caused by

15 flooding and of the necessity of increasing flood preparedness in both public administrations and population.

~~In the following sections, the most important assumptions of the methodology concerning the spatial resolution, the hydrological conditions, the breach locations and modelling are discussed thoroughly. For each topic, both general guidelines and specific assumptions for the pilot area are provided. Moreover, the most relevant simulation outputs and their usefulness for civil protection purposes are described.~~

20 **3.1  Setup**

The computational domain was built based on the available 1 m-resolution DTM of the riverbeds and of the floodable lowlands derived from LiDAR surveys carried out in the years 2008, 2015 and 2016. In

25 ~~traditional 2D and 1D-2D modelling. To date, however, both the parallelization of 2D codes (see Sect. 2.3) and the development of new remote sensing techniques, such as Light Detection and Ranging (LiDAR) and Shuttle Radar Topography Mission (SRTM), which provide raw data for digital terrain model (DTM) generation, allow the modellers to perform high resolution simulations for large areas. A fine mesh is often necessary to describe all the relevant terrain features typical of man-made landscape (e.g. roads, railways, channels, embankments) in detail.~~

30

~~Nowadays, high resolution DTMs of most flood-prone areas are available to public access, representing a powerful tool for accurate flood modelling. In particular, the whole study area is covered by LiDAR surveys carried out in the years 2008, 2015 and 2016. The bathymetry here adopted was hence built based on the available 1-m resolution DTMs of the riverbeds and of the floodable lowlands. However, in~~ order to avoid the excessive memory requirements and heavy computational costs (even for a fast GPU-parallel model), related to the adoption of a uniform 1 m mesh (which would require billions of cells), the DTM was downsampled to a resolution of 5 m. This operation did not affect the crest elevation of the artificial embankments. In fact, each 5×5 m cell crossed by an embankment was identified, and its elevation was set equal to the maximum value among the original 25 points belonging to that cell; otherwise, its elevation was simply computed as the average of the 25 terrain data comprised in that cell. For other urban features that were not captured correctly  by the LiDAR survey, additional corrections were introduced manually.

Then, the study domain was discretized by means of ~~computational cells and thus the runtimes (still retaining the same level of accuracy), the adoption of non-uniform meshes, both unstructured (Liang et al., 2008; Saetra et al., 2015) and structured (Vacondio et al., 2017), should be taken into consideration. In urban and suburban areas, the presence of road and railway embankments can influence the flood dynamics significantly, and the bathymetry near these elements should be at high resolution. On the other hand, for uniform rural areas a lower resolution can be used without impairing the overall accuracy. Therefore,with the finest resolution equal to 5 m was adopted in this work. The, which is shown in, consists of roughly 13·10⁶ cells, and the number of cells is reduced by approximately 70% compared with a uniform 5 m-mesh. This~~), whose spatial resolution is considered suitable for the detailed modelling of the river and the lowland area, consists of roughly $13\cdot10^6$ cells, and the number of cells is reduced by approximately 70% compared to a uniform 5 m-mesh.

The study area also includes several urban settlements: Modena, with about 185,000 inhabitants, is the most populated one, and its old city centre comprises narrow streets, which cannot be accurately described with a 5 m resolution. Therefore, limited to the few scenarios concerning the flooding of Modena, simulations were performed using a finer mesh (up to 2.5 m) in the town , and  the urban layout was modelled according to the "building hole" method (Schubert and Sanders, 2012), in order to capture the flow field inside the urban area correctly.

Buildings were explicitly resolved only for the city of Modena, whereas, according to previous findings (Vacondio et al., 2016), the presence of the other (smaller) settlements was taken into account by means of a higher resistance parameter ("building resistance" method, Schubert and Sanders, 2012). In particular, the roughness coefficient for the urban areas was calibrated based on the event occurred in 2014 (the flood arrival times at selected locations were known), and was set equal to 0.14 $m^{-1/3}$ s, while for rural areas a Manning coefficient of 0.05 $m^{-1/3}$ s was chosen. In the absence of data for calibration concerning past flooding events, land use maps can be exploited to assign standard roughness values from the literature.

As for the river roughness, the calibration for the Secchia River was performed in previous works (Vacondio et al., 2016), based on the water levels recorded at the available gauging stations during recent flood events. A Manning coefficient equal to 0.05 m$^{-1/3}$ s provided the best results. The roughness of the Panaro River was also subject to calibration with a similar procedure, and the value 0.04 m$^{-1/3}$ s was selected.

**scenarios**

As regards the selection of the hydrological

 for  the two

10 ~~scenarios were modelled, but the simulation database can be extended with more hydrological inputs, if needed. The first case ("inflow A") corresponds to the condition for which the water surface elevation reaches the levee crown somewhere along the river, thus generating overtopping. The second configuration ("inflow B") concerns a flood event with a lower return period, for which the levee is never overtopped, but other mechanisms might induce the levee collapse. In fact, earthen levees can also experience breaching for piping and internal erosion processes, even when water levels remain below~~

15 ~~the levee crown. Besides, the dens of burrowing animals (e.g. porcupine, badger, nutria) were recently identified as another possible cause for breach triggering (Orlandini et al., 2015). Incidentally, the collapse of an embankment during a flood event with a relatively low return period can be very threatening for human lives because the highest warning thresholds may not be reached, and population can be unprepared to face flooding.~~

20  rivers , the Synthetic Design Hydrographs (Tomirotti and Mignosa, 2017) with assigned return periods were considered. After preliminary simulations , inflow A (overtopping-induced flooding) was identified as the 50

25 years-return period hydrograph for the Secchia River, and as the 100 years-one for the Panaro River. Then, the inflow hydrograph with 20 years-return period was selected as inflow B for both rivers, in order to consider an event with  higher frequency. These discharge hydrographs were assumed as upstream boundary conditions during the simulations.

30

single-valued rating curve is imposed as outflow boundary condition, it should be assigned at the farthest possible section downstream from the breach location. For all the considered scenarios, theThe downstream section was located at the confluence (of the Secchia and Panaro rivers, respectively) into the Po River, and a constant water level in this (much larger) river, which did not affect the breach outflow even for the most downstream breach location, was assumed as boundary condition.

**3.3 Levee breaches location and modelling**

Several breach locations were identified along the twoAs mentioned before, the pilot area can be flooded by hypothetical failures occurring along the Secchia and Panaro rivers, so that a possible actual event can be associated to the closest simulated scenario. The pitch between two consecutive breach positions should be selected considering the river dimensions, the presence of urban settlements, the flood prone area topography, and the possible presence of roads and embankments influencing the inundation dynamics. In the present study, the breach locations were assumed at. Assuming a distance of about 2 km from each one breach to the other, approximately. As a result, 30 breach positionslocations were selected onidentified along the right levee of the Secchia River (for the flooding scenarios related to this river), whileand 26 were identifiedones along the left levee of the Panaro River. Figure 1 reports all the 56 simulated breach sites. Among these, 8 breach scenarios on the Secchia River and 4 on the Panaro River involve the city of Modena. Based on past observations, the breach final width was assumed equal to 100 m for all scenarios, while the opening time was set equal to 3 or 6 h for inflows A and B, respectively.

The hypothesis of instantaneous break is not realistic for river embankments; hence,All the description of the gradual opening of the levee breach must be somehow included in the 2D modelling. Among the available approaches in the literature, which also include the coupling of the SWEs with a sediment transport model (Faeh, 2007), or with an erosion law (Dazzi et al., 2019), the "geometric" approach is selected in this work. The breach opening is described by varying the local topography in time, assuming a trapezoidal shape and imposing the breach geometric dimensions and failure time as input parameters. This method was successfully applied to a real test case (Vacondio et al., 2016). A similar approach is often adopted for 1D-2D coupled models, especially in the context of flood hazard assessment (e.g. Vorogushyn et al., 2010; Mazzoleni et al., 2014). Given the uncertainties about the geotechnical parameters of the embankment and the complexity of the breaching process (three dimensionality, interactions between erosion, infiltration, and bank stability, etc.), this simple "geometric" approach can be considered adequate for the purpose of this study.

The breach model parameters should be consistent with historical data, if available (e.g. Nagy, 2006; Vorogushyn et al., 2010; Govi and Maraga, 2005), or otherwise they should be identified according to the river and embankment characteristics. A breach final width in the order of tens to hundreds of meters is often assumed in other works (Apel et al., 2006; Kamrath et al., 2006), and the uncertainty in its value is sometimes considered with a probabilistic treatment (Apel et al., 2006; Vorogushyn et al., 2010; Mazzoleni et al., 2014) or a sensitivity analysis (Kamrath et al., 2006; Huthoff et al., 2015). As for the breach development time, very few field observations are available, and often values in the range 1-3 h are assumed112

simulations (56 breach locations and 2 hydrological scenarios)

5

10  were prolonged for 72 h (3 days), because at that point the outflow from the breach was almost null, the flooded area had reached its maximum extension, and no significant  flow dynamics can be observed.

**3.2 Outcomes**

15 ~~In this Section, some results of the application of the RESILIENCE methodology to the pilot area are presented. Since 56 breach locations and 2 hydrological scenarios were considered, the database for this area currently includes the results of 112 simulations. As already discussed, the outcomes of these scenarios could help the civil protection activities for emergency planning and/or at the occurrence of an event similar to one of those already modelled. Arrighi et al. (2019) recently presented a framework that integrates hydrologic/hydraulic modelling with human safety and transport accessibility~~

20 ~~evaluations, applied to a small municipality near Florence (Italy), and confirmed the usefulness of detailed spatial and temporal flood data provided by numerical modelling for civil protection purposes. Thus, the first mandatory output concerns spatial and temporal information about the flood dynamics in the lowland area, which can be visualized as an animation of the inundation pattern or as a sequence of frames at selected times.An example of video showing the flooding evolution for one scenario on the Secchia River will be provided as additional material.~~

[revised manuscript text omitted]

**Table 1. Sensitivity analysis on the final breach width *L* and opening time *T* for one scenario on the Secchia River (see Fig. 3), with both inflows A and B. Results are compared based on the total outflow volume *vol*, and the flooded area 24, 48 and 72 h after the breach opening (*area₂₄*, *area₄₈* and *area₇₂*, respectively). Their relative differences ($\Delta vol$, $\Delta area_{24}$, $\Delta area_{48}$, $\Delta area_{72}$) with reference to the baseline simulation are also reported for each tested configuration.**